# Risk of Anxiety and Depression in Patients with Inflammatory Bowel Disease: A Nationwide, Population-Based Study

**DOI:** 10.3390/jcm8050654

**Published:** 2019-05-10

**Authors:** Kookhwan Choi, Jaeyoung Chun, Kyungdo Han, Seona Park, Hosim Soh, Jihye Kim, Jooyoung Lee, Hyun Jung Lee, Jong Pil Im, Joo Sung Kim

**Affiliations:** 1Department of Internal Medicine and Liver Research Institute, Seoul National University College of Medicine, 101 Daehak-ro, Chongno-gu, Seoul 110-799, Korea; kook0104@naver.com (K.C.); j40479@gmail.com (J.C.); seven0526@naver.com (S.P.); hosimsoh@gmail.com (H.S.); hyehye08@gmail.com (J.K.); cheeryoung@hanmail.net (J.L.); guswjd80@gmail.com (H.J.L.); jp-im@hanmail.net (J.P.I.); 2Department of Internal Medicine, Gangnam Severance Hospital, Yonsei University College of Medicine, Seoul 06273, Korea; 3Department of Medical Statistics, College of Medicine, The Catholic University of Korea, Seoul 06591, Korea; hkd917@naver.com

**Keywords:** anxiety, claims data, depression, inflammatory bowel disease

## Abstract

Background and Aims: Inflammatory bowel disease (IBD) may be associated with anxiety and depression. The aim of this study was to evaluate the incidence of anxiety and depression in patients with IBD compared to the general population. Methods: A nationwide population-based cohort study was conducted using claims data from the National Healthcare Insurance service in Korea. We compared the incidence of anxiety and depression between 15,569 IBD patients and 46,707 non-IBD controls, age and sex matched at a ratio of 1:3. Results: During a mean follow-up of six years, IBD patients experienced significantly more anxiety (12.2% vs. 8.7%; *p* < 0.001) and depression (8.0% vs. 4.7%; *p* < 0.001) compared to controls. The curves showing cumulative incidences of anxiety and depression showed a steep rise within one year following a diagnosis of IBD, leading to lines with a constant slope. The hazard ratio (HR) for new onset anxiety following a diagnosis of Crohn’s disease (CD) and ulcerative colitis (UC) was 1.63 and 1.60, respectively, compared to controls (*p* < 0.001). Compared to controls, the HR for developing depression after a diagnosis of CD and UC was 2.09 and 2.00, respectively (*p* < 0.001). The risks of anxiety and depression in patients with IBD were higher compared to controls, except in those with diabetes mellitus, hypertension, and dyslipidemia, or who required immunomodulators and biologics within one year of the IBD diagnosis. Conclusions: The risk of anxiety and depression increased after a diagnosis of IBD compared to the general population.

## 1. Introduction 

South Korea has one of the worst mental health problems among industrialized countries. The prevalence of depression in Korean population is similar to western countries [1], but the suicide rate in South Korea has rapidly increased in a recent decade [2], up to the highest level among the Organized for Economic Cooperation and Development countries. The prevalence of anxiety in the Korean population is also higher compared to other East Asian countries [3]. All the reasons for the Korean population being exposed to potential risks of anxiety and depression have not been revealed, but it is customary for them to work or study late into the night in stressful environments.

Inflammatory bowel disease (IBD), including Crohn’s disease (CD) and ulcerative colitis (UC), is a chronic and relapsing inflammatory disorder primarily involving the gastrointestinal tract [4]. In a recent epidemiologic study, the incidence and prevalence of UC and CD have increased in South Korea [5,6]. Accumulating evidence suggests that complex interactions between genetic and environmental factors are related to the initiation and perpetuation of intestinal inflammation [7,8]. Alterations in brain-gut interactions also contribute to the pathogenesis of IBD [9]. The complex communication system between the brain and the gastrointestinal tract involves modulating interactions among the autonomic nervous system, the hypothalamic-pituitary-adrenal (HPA) axis, the gastrointestinal immune system, and intestinal microbiota [10]. A dysregulated brain-gut axis may lead to psychological comorbidities, as well as an exacerbation of gastrointestinal inflammation under stressful conditions in patients with IBD.

Anxiety and depression are the most common psychological disorders in patients with IBD and patient-reported outcome measures (PROMs) for anxiety and depression are considered to be critical to IBD patient care [11,12]. Indeed, IBD is strongly related to comorbid anxiety and depression [13,14,15,16,17]. A recent population-based study based in Canada reported that the incidence of anxiety and depression significantly increased in patients with immune-mediated inflammatory diseases (IMIDs), including IBD, multiple sclerosis, and rheumatoid arthritis, compared to the general population as early as five years prior to the diagnosis of IMIDs [18]. Studies have previously reported that the occurrence of anxiety and depression is highly associated with IBD. A population-based study reported that the lifetime prevalence of anxiety and depression in patients with IBD were 24.4–31.9% and 21.8–22.5%, respectively [19]. In a recent systematic review for anxiety and depression comorbid with IBD, the pooled prevalence of anxiety and depression were reported to be 19.1% and 21.2% in patients with IBD, respectively [20]. Assessing the incidence and risk of anxiety and depression in patients with IBD is clinically important because anxiety and depression have been identified as factors exacerbating the disease course of IBD [21]. However, population-based estimates regarding the risk for the newly developed anxiety and depression following a diagnosis of IBD have not been fully determined yet. The aim of this study, therefore, is to investigate the incidence and risk factors of anxiety and depression in patients with IBD through a nationwide population-based study.

## 2. Materials and Methods

### 2.1. Data Source and Patient Selection 

The National Healthcare Insurance Service (NHIS) database was used for this population-based cohort study. The NHIS is the nationwide comprehensive compulsory health care system which covers approximately 97% of the population in South Korea, totaling over 51 million people [22], the remaining 3% of which is covered by Medical Aid. The NHIS database includes information regarding demographics, outpatient and inpatient care, disease diagnosis, prescriptions, and procedure records. The NHIS database represents individuals with non-identifiable codes in order to protect their personal information [23]. In 2006, the NHIS established a registration program for rare intractable diseases (RIDs), such as CD and UC and patients with IBD must register in the RID program to receive their co-payment reduction of up to 90% in Korea. To qualify for enrollment in this special co-payment program, patients must meet the diagnostic criteria for each RID as described by the NHIS and be certified by specialized physicians. The diagnostic codes were defined by the International Classification of Diseases, tenth revision (ICD-10) code and a special code (V code) registered in the RID database [6]. Patients with IBD are required to meet the clinical, endoscopic, and histological findings diagnostic criteria to be enrolled in the RID registration program. 

From January 2010 to December 2013, all patients with IBD assigned by both the ICD-10 and V codes were included in this study. The patients with CD were identified by ICD-10 (K50) and V codes (V130) and those with UC were detected using ICD-10 (K51) and V codes (V131). All patients with IBD who had a past history of anxiety and depression were excluded in order to only include individuals who experienced an initial onset of anxiety and/or depression following the presentation of IBD. The study participants were divided into the following two groups: An ‘incident group’, defined as patients who were newly diagnosed with IBD between January 2010 and December 2013; and a ‘prevalent’ group defined as those who had been previously diagnosed with IBD and were also coded with IBD between January 2010 and December 2013. For the identification of incident cases of IBD, we included a washout period of six years (from January 2005 to December 2010). Patients who were diagnosed with IBD before January 2010 but were not coded from 2010 to 2013 were excluded from this study. Three individuals, matched by age and sex, who were not diagnosed with IBD were randomly selected for each IBD patient and enrolled in the non-IBD control group for this study. All subjects were East Asian.

To evaluate the diagnostic accuracy for CD and UC using both ICD-10 and V codes, a sensitivity analysis was conducted by a retrospective review of medical records of IBD patients at Seoul National University Hospital (SNUH), a tertiary referral hospital in South Korea, from January 2010 to December 2013, as previously described [24]. The sensitivities for the detection of CD and UC were 94.5% and 96.4%, respectively. 

### 2.2. Data Collection and Study Outcomes

Demographic data of the study population (age, sex, residence, and income), comorbidities, and medication use for IBD were collected. Comorbidities, including hypertension (ICD-10 code: I10-13, and I15, and medication uses for treatment of hypertension), DM (E11-14 and medication uses for treatment of DM), dyslipidemia (E78 and medication uses for treatment of dyslipidemia), congestive heart failure (I50), ischemic heart disease (I20-I25), cerebrovascular disease (I63-64), chronic obstructive pulmonary disease (COPD; J41-44), end-stage renal disease (ESRD; N18-19, Z49, Z94.0 and Z99.2, and requirement for hemodialysis) and malignancy (C00-96) that were defined using ICD-10 codes were collected, as previously described [24,25,26,27,28]. We also collected information about therapeutic drug use for IBD, including immunomodulators (azathioprine, 6-mercaptopurine, and methotrexate), steroids, and biologics (infliximab and adalimumab) within one year of IBD diagnosis. 

The primary outcomes analyzed were newly diagnosed with anxiety and depression. Anxiety and depression were detected using ICD-10 codes as previously defined; F40-42 for anxiety and F32-34 for depression [19,29,30,31]. During the follow-up, patients without newly developed anxiety and depression were censored on the last day of the follow-up or the date of death. 

### 2.3. Statistical Analysis

Continuous variables are presented as means ± standard deviation (SD) and categorical variables are presented as number and percentage. To compare characteristics between IBD and control groups, t-tests were used for continuous variables and chi-square tests were used for binary and categorical variables. The cumulative anxiety and depression incidence for each group was plotted with Kaplan-Meier curves and compared using the log-rank test. Cox regression models were used to assess the risks of newly developed anxiety and depression associated with baseline characteristics, comorbid medical conditions, and therapeutic drug use for IBD. The potential effect modification by age, sex, income, residence, comorbid medical conditions, and therapeutic drug use for IBD was evaluated through stratified analysis and interaction testing using a forest plot. All statistical tests were two-tailed and the significance level was set at a *p* < 0.05. Statistical analyses were performed using R programming version 3.4.3 (The R Foundation for Statistical Computing, Vienna, Austria, http://www.R-project.org) and SAS Version 9.2 (SAS Institute Inc., Cary, NC, USA) for Windows. The study protocol was reviewed and approved by the Institutional Review Board of Seoul National University Hospital (SNUH IRB No. H-1703-107-840).

## 3. Results 

### 3.1. Baseline Characteristics of the Study Population

From January 2010 to December 2013, a total of 15,569 IBD patients and 46,707 age- and sex-matched controls were enrolled in this study. The baseline characteristics of the study population are shown in Table 1. Mean age of the study population was 32.0 years, with 11,409 males (73.3%) in patients with IBD (78.3% in CD and 69.8% in UC, respectively). The IBD group showed a significantly higher proportion of urban residence and a higher income compared to non-IBD controls (*p* < 0.001). The prevalence of DM was significantly higher in the IBD group compared to non-IBD controls (*p* = 0.035). Interestingly, the UC but not the CD group had a significantly higher prevalence of DM compared to non-IBD controls (*p* = 0.016). In contrast, the CD group showed a significantly higher prevalence of hypertension compared to non-IBD controls (*p* = 0.036). The IBD group showed a significantly higher prevalence of congestive heart failure, ischemic heart disease, COPD, ESRD and malignancy compared to non-IBD controls (*p* < 0.001 for each variable). Finally, immunomodulators, steroids, and biologics were used significantly more frequently in both IBD groups, compared to non-IBD controls (*p* < 0.001 for each drug). 

### 3.2. Incidence and Risk of Anxiety and Depression in IBD 

During a mean follow-up of six years, the incidence of anxiety in patients with IBD was significantly higher compared to non-IBD controls (12.2% vs. 8.7%; *p <* 0.001). In patients with CD, cumulative incidences of anxiety were 3.0%, 6.9%, and 11.5% at one, three, and six years after the diagnosis of CD, respectively. For UC, cumulative incidences of anxiety were 4.2%, 9.9%, and 16.7% in one, three, and six years after the diagnosis of UC, respectively (see Figure 1A–C).

Similarly, during the follow-up, the incidence of depression in patients with IBD was significantly higher compared to non-IBD controls (8.0% vs. 3.7%; *p* < 0.001). In patients with CD, cumulative incidences of depression were 2.7%, 5.2%, and 8.0% in one, three, and six years after the diagnosis of CD, respectively. In those with UC, cumulative incidences of depression were 2.6%, 6.6%, and 10.8% in one, three, and six years after the diagnosis of UC, respectively (see Figure 1D–F). 

After the adjustment for confounding factors, including age, sex, residence, income, and comorbid medical conditions, the Cox proportional hazard models showed the comparative risks of newly developed anxiety and depression between IBD and non-IBD control groups using an adjusted hazard ratio (HR) with 95% confidence intervals (CI) in Table 2 and Table 3. In the CD incident group, the incidence rate (per 1000 person-years) of anxiety was 20.88, compared to 14.31 in non-CD controls (adjusted HR, 1.58; 95% CI, 1.38–1.82; *p* < 0.001), and the incidence rate of depression was 14.99, compared to 7.75 in non-CD controls (adjusted HR, 2.06; 95% CI, 1.74–2.44; *p* < 0.001). The incidence rate (per 1000 person-years) of anxiety was 31.19 in the UC incident group, compared to 21.55 in non-UC controls (adjusted HR, 1.58; 95% CI, 1.43–1.74; *p* < 0.001), and the incidence rate of depression was 19.63 in the UC incident group, compared to 11.28 per in non-UC controls (adjusted HR, 1.93; 95% CI, 1.70–2.18; *p* < 0.001).

### 3.3. Subgroup Analyses

All subgroups of patients with CD, except those with dyslipidemia and hypertension and concurrent medication with immunomodulators and biologics, were at an increased risk of anxiety compared to non-CD controls. The impact of CD on the risk of anxiety was more pronounced in patients who were not taking immunomodulators (adjusted HR, 1.90 vs. 0.56; interaction *p* = 0.0125). Additionally, all subgroups had an increased risk of anxiety compared to non-UC controls, except patients with immunomodulator and biologic therapy. In addition, the impact of UC on the development of anxiety was less pronounced in patients with steroids therapy (adjusted HR, 1.41 vs. 1.68; interaction *p* = 0.0476) or comorbid medical conditions (adjusted HR, 1.40 vs. 1.75; interaction *p =* 0.0446) (see Figure 2).

All CD patients were at greater risk of depression compared to non-CD controls, except those who were diagnosed with dyslipidemia or prescribed immunomodulator and biologic therapy. The impact of CD on the presentation of depression significantly increased in male patients compared to female (adjusted HR, 1.58 vs. 1.21; interaction *p* = 0.0250). Additionally, all UC subgroups had an increased risk of depression compared to non-UC controls, except patients with comorbid medical conditions, particularly those with DM and dyslipidemia, or those who required immunomodulators and biologics for IBD. The impact of UC on the development of depression was more pronounced in patients without comorbid medical conditions (adjusted HR, 1.53 vs. 1.13; interaction *p* = 0.0007), particularly in those with hypertension (adjusted HR, 1.49 vs. 1.17; interaction *p* = 0.0182) and dyslipidemia (adjusted HR, 1.49 vs. 1.09; interaction *p* = 0.0271). In UC patients with immunomodulator therapy, the impact of UC on developing depression significantly decreased compared to those who did not use immunomodulators (adjusted HR, 1.45 vs. 0.68; interaction *p* = 0.0056) (see Figure 3).

### 3.4. Risk of Anxiety and Depression in IBD Based on Medication Use 

Among the IBD patients, Cox proportional hazard models showed the comparative risks of newly developed anxiety and depression according to medication use for IBD, using an adjusted hazard ratio (HR) with 95% confidence intervals (CI) in Appendix A. Both UC and CD patients who were exposed to steroids within one year after the diagnosis had a significantly additive risk of comorbid anxiety and depression compared to those without steroid therapy. IBD patients who received biologics within one year after the diagnosis had a significantly increased risk of developing depression, but not anxiety, compared to those without biologics.

## 4. Discussion

The results from this nationwide, population-based cohort study utilizing approximately 62,000 individuals in the NHIS database suggest a significantly increased risk for developing anxiety and depression in patients with IBD compared to the general population. To the best of our knowledge, this is the largest population-based cohort study using claims data to elucidate the risk of initial onset anxiety and depression related to the diagnosis and treatment of IBD. The incidence rates of anxiety and depression in the CD incident group were 19.5 and 12.8 per 1000 person-years, respectively. Additionally, in the UC incident group, the incidence rates of anxiety and depression were 28.9 and 16.5 per 1000 person-years, respectively. The adjusted risks of new onset anxiety and depression following the diagnosis of IBD were significantly higher at 1.6- and 2.0-fold, respectively, when compared to the general population.

These findings are consistent with our results, demonstrating that cumulative incidences of anxiety and depression at six years following the diagnosis of IBD were 11.5% and 8.0%, respectively, in CD and 16.7% and 10.8%, respectively, in UC, with a continuously increasing trend of linearity. Interestingly, the curves representing the cumulative incidences of anxiety and depression indicated a steep rise within one year following the diagnosis of IBD, leading to lines with a constant slope. This ‘steep rise and then steady increase’ phenomenon was also observed in a nested case-control study from Southern England, showing the highest risk for anxiety and depression within one year following the diagnosis of IBD [32]. There are several possible explanations for the phenomenon of the development of anxiety and depression in patients with IBD. For example, the initiation of anxiety and depression may be linked to the presentation of active intestinal inflammation related to the dysregulations of the brain-gut axis. Furthermore, psychological stress on patients’ awareness of the diagnosis of chronic relapsing disorders may make them seriously anxious or depressed. Finally, some patients may have already had anxiety or depression but were diagnosed with both IBD and these psychological comorbidities simultaneously by the physicians at one of the first visits to the clinics. The important thing is that the incidences of anxiety and depression remained constant after the ‘steep rise’, leading to the high cumulative incidence of psychological comorbidities of up to 17% at six years after the diagnosis of IBD. Therefore, the potential risk for the occurrence of anxiety and depression should be recognized and monitored carefully in patients with IBD, especially within one year after the diagnosis of IBD based on the ‘steep rise and then steady increase’ phenomenon. 

In general, the risks of anxiety and depression in those with IBD were higher compared to the general population in this study. The relative risk for newly developed anxiety and depression related to IBD was generally subdued in patients with comorbid medical conditions or those who required early immunomodulatory and biologic therapy within one year following their IBD diagnosis. However, newly diagnosed IBD patients were at significant adjusted comparative risks of developing anxiety and depression in spite of a high proportion of comorbid medical conditions or immunosuppressive therapy, compared to non-IBD controls. Previous studies have shown that patients with comorbid chronic medical conditions, including metabolic syndrome, cardiovascular diseases, stroke, COPD, ESRD, and malignancy, which cause psychological stress are at an increased risk of anxiety and depression [33,34,35,36,37,38]. Patients who needed immunomodulatory and biologic therapy for the treatment of IBD also had an increased risk for comorbid anxiety and depression related to the disease severity of IBD [39]. These findings suggest that the presence of comorbid medical conditions or concurrent medication with immunomodulatory and biologics are independent risk factors for anxiety and depression. Individuals at low risk of anxiety and depression seem to be exposed to the risk of developing anxiety and depression, due to the diagnosis of IBD. Thus, IBD itself may increase the risk of developing anxiety and depression based on these results. 

The impact of CD on the development of depression was significantly more pronounced in male patients than in females. The significant impact of UC on the risk of depression was also noted, regardless of gender. Previous studies demonstrated a significantly higher heritability of major depressive disorders in females, which may be related to a higher prevalence of depression in women [40]. Therefore, male individuals in the non-IBD control group were at relatively low risk of depression among the study population, but IBD itself, both UC and CD may increase the risk of depression in male patients. In contrast, the women with UC with the inherited risk for depression might have an additive risk compared to the general population, although the pathophysiology remains elusive.

In contrast, the influence of IBD on the occurrence of anxiety and depression was seen irrespective of steroid use, even though the impact of UC on initial onset anxiety was more pronounced in patients without steroid therapy. This is consistent with the results of a recent prospective study reporting that disease activity of IBD at diagnosis was associated with a 6-fold increase in risk for developing anxiety [3]. Among the patients with quiescent IBD, anxiety at diagnosis was also significantly related to disease exacerbation of IBD [3]. Taken together, active inflammation in the gastrointestinal tract may add to mood disorders, including anxiety and depression, with bidirectional effects of gut inflammation and psychological disorders. 

IBD patients have an additional burden of psychological stress associated with chronic and relapsing intestinal inflammation. Chronic medical illness may cause adverse psychiatric effects and mental problems [41,42]. Immunoregulatory pathways related to the pathogenesis of inflammatory diseases may influence the development of psychological disorders. Neuropeptides which are involved in the gut-brain communication have a critical role in this pathway [43]. Interestingly, the presence of psychological diseases may predispose to developing IBD [21,44], although it remains elusive. In a population-based cohort study from Canada, nearly 80% of patients with IBD and anxiety were diagnosed with anxiety disorder more than 2 years prior to the diagnosis of IBD [45]. Depression may lead to changes in immunosuppression and inflammation of the gastrointestinal tract [46]. Psychological stress-induced proinflammatory cytokines may cause destruction of intestinal mucosa and microbiota [47]. In a multicenter cohort study from the United State, comorbid anxiety or depression was associated with a 28% increase in surgical risk in CD, suggesting the influence of psychological comorbidities on the disease-related outcome [48]. Taken together, mood disorders and IBD may have a reciprocal influence on the initiation and progression of the diseases. 

This study has several limitations, due to its retrospective nature. First, we could not analyze the influence of IBD on the potential risk of depression and anxiety according to the severity of intestinal inflammation, because the disease severity of IBD was not available from the NHIS database. Among the IBD patients, the increased risks of comorbid anxiety and depression in those receiving steroids and biologics, except the anxiety risk in biologics users, might reflect the effects of severe disease activity leading to use of steroids and biologics on the development of these psychological comorbidities. Further investigations are required to determine the impacts of medication uses for IBD on the risks of psychological comorbidities considering disease activity of IBD. Second, there were differences in the baseline covariables between IBD and non-IBD groups, except age and sex, because the non-IBD controls were randomly selected for each IBD patient matched by age and sex. It was difficult to control the effects of all covariables on the study outcomes over time by the matching in time-dependent Cox–proportional hazard models. Therefore, multivariable Cox regression models with adjustment for all covariables were used to minimize their confounding effects on the development of anxiety and depression. Third, the standardized PROMs for the identification of anxiety and depression were not available in this study. The operational definitions of anxiety and depression used in this study has not been validated yet in South Korea. As previously described [31], the incidence of anxiety and depression in IBD might be underdiagnosed than found by screening tools or standardized PROMs for mood disorders. However, the comparative risk of anxiety and depression between IBD and non-IBD groups might not have been affected by the validity of the operational definitions because of the same diagnostic criteria applied between the two groups. Finally, some of the IBD patients who were diagnosed with anxiety and depression after the diagnosis of IBD might have psychological manifestations before the diagnosis among the study population. In addition, it is very difficult to differentiate between these psychological comorbidities and natural stress response at the diagnosis of IBD in this study. It may have contributed to the early “steep rise” phenomenon of IBD patients in real practice. Therefore, further large-scale prospective studies using standard diagnostic tools should define the onset and incidence of comorbid anxiety and depression in IBD, which reflects population-based estimates in real practice. 

In conclusion, the risk of anxiety and depression in patients with IBD was significantly higher compared to the general population. Therefore, physicians should be aware of the potential risk for anxiety and depression following the diagnosis of IBD in practice, especially within one year following the diagnosis of IBD, considering the greatest risks of developing these psychological comorbidities and their potential effects on the disease course of IBD.

## Figures and Tables

**Figure 1 jcm-08-00654-f001:**
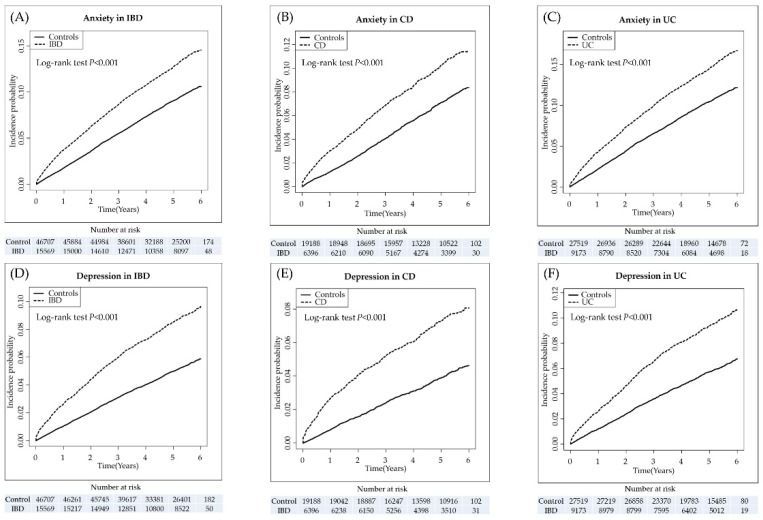
Cumulative incidence of anxiety and depression. Kaplan-Meier curves showed the incidence of anxiety (**A**–**C**) in patients with inflammatory bowel disease (**A**), Crohn’s disease (**B**), and ulcerative colitis (**C**), and depression (D-F) in those with inflammatory bowel disease (**D**), Crohn’s disease (**E**), and ulcerative colitis (**F**), compared to the general population, respectively. CD, Crohn’s disease; IBD, inflammatory bowel disease; UC, ulcerative colitis.

**Figure 2 jcm-08-00654-f002:**
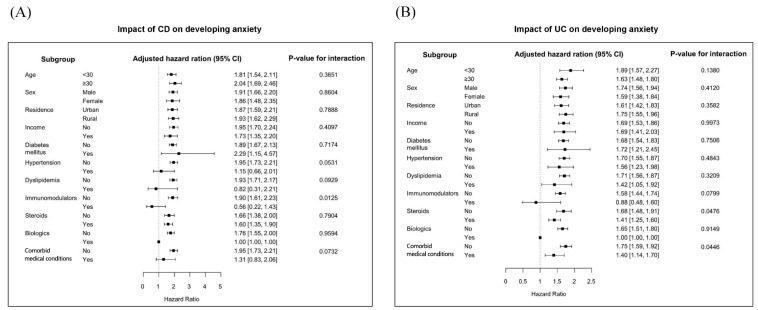
Subgroup analysis for the risk of anxiety in patients with (**A**) Crohn’s disease; and (**B**) ulcerative colitis compared to control groups. CD, Crohn’s disease; CI, confidence interval; UC, ulcerative colitis.

**Figure 3 jcm-08-00654-f003:**
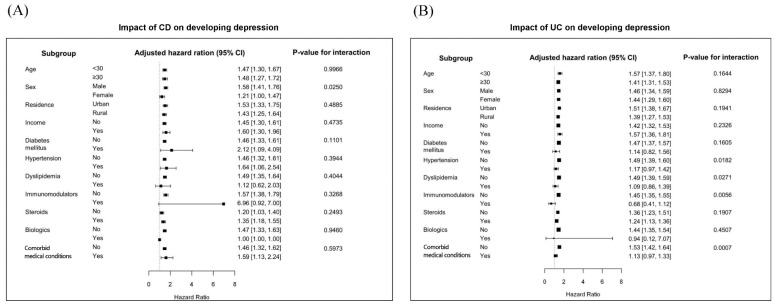
Subgroup analysis for the risk of depression in patients with (**A**) Crohn’s disease; and (**B**) ulcerative colitis compared to control groups. CD, Crohn’s disease; CI, confidence interval; UC, ulcerative colitis.

**Table 1 jcm-08-00654-t001:** Baseline characteristics of the study population.

No. (%)	IBD			CD			UC		
	Controls ^‡^	IBD	*p*-Value	Controls	CD	*p*-Value	Controls	UC	*p*-Value
Events	46,707	15,569		19,188	6396		27,519	9173	
Age, years ^†^	32.0 ± 13.9	32.0 ± 13.9	1	25.3 ± 10.7	25.3 ± 10.7	1	36.7 ± 13.9	36.7 ± 13.9	1
<15	3054 (6.5)	1018 (6.5)		2145 (11.2)	715 (11.2)		909 (3.3)	303 (3.3)	
15–29	25,323 (54.2)	8441 (54.2)		13,611 (70.9)	4537 (70.9)		11,712 (42.6)	3904 (42.6)	
30–44	17,505(37.5)	5835 (37.5)		3363 (17.5)	1121 (17.5)		14,142 (51.4)	4714 (51.4)	
>45	825 (1.8)	275 (1.8)		69 (0.4)	23 (0.4)		756 (2.7)	252 (2.7)	
Gender: Male	34,227 (73.3)	11,409 (73.3)	1	15,027 (78.3)	5009 (78.3)	1	19,200 (69.8)	6400 (69.8)	1
Residence			<0.001			<0.001			<0.001
Urban	21,924 (46.9)	8125 (52.2)		9003 (46.9)	3398 (53.1)		12,921 (46.9)	4727 (51.5)	
Rural	24,783 (53.1)	7444 (47.8)		10,185 (53.1)	2998 (46.9)		14,598 (53.0)	4446 (48.5)	
Income *			<0.001			<0.001			<0.001
Q2-4	36,284 (77.7)	12,678 (81.4)		14,846 (77.4)	5123 (80.1)		21,438 (77.9)	7555 (82.4)	
Q1	10,423 (22.3)	2891 (18.6)		4342 (22.6)	1273 (19.9)		6081 (22.1)	1618 (17.6)	
**Comorbid medical conditions**									
Diabetes mellitus	1048 (2.2)	305 (2.0)	0.035	156 (0.8)	54 (0.8)	0.810	892 (3.2)	251 (2.7)	0.016
Hypertension	2660 (5.7)	822 (5.3)	0.051	429 (2.2)	115 (1.8)	0.036	2231 (8.1)	707 (7.7)	0.222
Dyslipidemia	1489 (3.2)	529 (3.4)	0.200	242 (1.3)	78 (1.2)	0.800	1247 (4.5)	451 (4.9)	0.128
Congestive heart failure	85 (0.2)	54 (0.4)	<0.001	14 (0.1)	12 (0.2)	0.013	71 (0.3)	42 (0.5)	0.003
Ischemic heart disease	581 (1.2)	339 (2.2)	<0.001	95 (0.5)	92 (1.4)	<0.001	486 (1.8)	247 (2.7)	<0.001
COPD	1,194 (2.6)	701 (4.5)	<0.001	385 (2.0)	325 (5.1)	<0.001	809 (2.9)	376 (4.1)	<0.001
Cerebrovascular disease	462 (1.0)	181 (1.2)	0.064	79 (0.4)	30 (0.5)	0.542	383 (1.4)	151 (1.7)	0.078
ESRD	41 (0.1)	40 (0.3)	<0.001	13 (0.1)	18 (0.3)	<0.001	28 (0.1)	22 (0.2)	0.002
Malignancy	355 (0.8)	179 (1.2)	<0.001	68 (0.4)	36 (0.6)	0.023	287 (1.0)	143 (1.6)	<0.001
**Use of therapeutic drugs**									
Immunomodulators	105 (0.2)	5194 (33.4)	<0.001	34 (0.2)	3865 (60.4)	<0.001	71 (0.3)	1329 (14.5)	<0.001
Steroids	12,701 (27.2)	8681 (55.8)	<0.001	4895 (25.5)	3689 (57.7)	<0.001	7806 (28.4)	4992 (54.4)	<0.001
Biologics	6 (0.01)	1122 (7.2)	<0.001	2 (0.01)	932 (14.6)	<0.001	4 (0.01)	190 (2.0)	<0.001

* Q1: Lower 25%, Q2-4: Upper 75%; ^†^ Mean ± SD; ^‡^ Controls: Age, Sex matched; CD, Crohn’s disease; COPD, Chronic obstructive pulmonary disease; ESRD, End-stage renal disease; IBD, Inflammatory bowel disease; No, Number; Q, Quartile; UC, Ulcerative colitis.

**Table 2 jcm-08-00654-t002:** Incidence and risk of anxiety in patients with inflammatory bowel disease compared to the general population.

	Events (n)	Follow-Up Duration (person-years)	Incidence Rate (per 1000 person-years)	Adjusted HR^*^ (95% CI)	*p*-Value
CD					
Controls	1301	90,912.81	14.31	1 (Ref)	
Prevalent	617	29,589.89	20.85	1.45 (1.32–1.60)	<0.001
Incident	236	11,300.82	20.88	1.58 (1.38–1.82)	<0.001
UC					
Controls	2779	128,958.37	21.55	1 (Ref)	
Prevalent	1287	41,764.84	30.82	1.44 (1.34–1.53)	<0.001
Incident	481	15,420.67	31.19	1.58 (1.43–1.74)	<0.001

CD, Crohn’s disease; CI, Confidence interval; HR, Hazard ratio; IBD, Inflammatory bowel disease; Ref, reference; UC, Ulcerative colitis. *Adjusted by age, sex, residence, income, and comorbid medical conditions, including diabetes mellitus, hypertension, dyslipidemia, congestive heart failure, ischemic heart disease, chronic pulmonary obstructive disease, cerebrovascular disease, end-stage renal disease, and malignancy.

**Table 3 jcm-08-00654-t003:** Incidence and risk of depression in patients with inflammatory bowel disease compared to the general population.

	Events (n)	Follow-Up Duration (person-years)	Incidence Rate (per 1000 person-years)	Adjusted HR^*^ (95% CI)	*p*-Value
CD					
Controls	716	92415.12	7.75	1 (Ref)	
Prevalent	437	30048.66	14.54	1.85 (1.64–2.09)	<0.001
Incident	171	10,864.02	14.99	2.06 (1.74–2.44)	<0.001
UC					
Controls	1495	132,419.52	11.28	1 (Ref)	
Prevalent	807	43,276.09	18.65	1.66 (1.52–1.81)	<0.001
Incident	312	15,890.80	19.63	1.93 (1.70–2.18)	<0.001

CD, Crohn’s disease; CI, Confidence interval; HR, Hazard ratio; IBD, Inflammatory bowel disease; Ref, reference; UC, Ulcerative colitis. *Adjusted by age, sex, residence, income, and comorbid medical conditions, including diabetes mellitus, hypertension, dyslipidemia, congestive heart failure, ischemic heart disease, chronic pulmonary obstructive disease, cerebrovascular disease, end-stage renal disease, and malignancy

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
