# Peer review of "Risk of Anxiety and Depression in Patients with Inflammatory Bowel Disease: A Nationwide, Population-Based Study"

_jcm, 2019, doi:10.3390/jcm8050654_

Round 1
Reviewer 1 Report
Thank you for allowing me to review the paper titled, "Risk of Anxiety and Depression in Patients with Inflammatory Bowel Disease: A Nationwide, Population-based Study". The aim of this study was to evaluate the incidence of anxiety and depression in patients with IBD compared to the general population. The authors report on a large nationwide population-based cohort study and use claims data from the National Healthcare Insurance service in Korea. They compared the incidence of anxiety and depression between 15,569 IBD patients and 46,707 non-IBD controls, age and sex matched at a ratio of 1:3. Overall this was a well written paper with several strengths, including a large sample size, registry data with high incentive to register, and examine UC and CD separately. This reviewer does have some suggestions to fine tune this strong manuscript. They will be reported by section below.
Introduction
General Comments: Overall the Introduction is well written. A few general questions, observations, comments listed below.
1. Typically the rate and incidence of anxiety and depression is presented in the Introduction. However, the authors report this information in the Discussion. I am not familiar with the format of this journal, but the authors may want to consider providing this information at the start of the paper. In addition, the impact and import of assessing the prevalence and incidence of anxiety is not presented in the introduction. This should be added.
2. Does the author have access regarding the ethnicity of this sample? This reviewer suspects that this national sample is mostly individuals of Asian decent. This should be added to the Methods if there is diversity in ethnicity, but in addition to this, are there cultural reasons why it is important to assess anxiety and depression in this population? (Also, for the discussion, are there cultural differences in reporting anxiety and depression based on gender in South Korea compared to western countries)?
3. Some of the references are outdated and would benefit from a larger literature search.
Methods
General Comments: Overall very well written and easy to read. The addition of sensitivity for the detection of UC and CD is excellent. A few general questions, observations, comments listed below.
1. It seems like the criteria for IBD is very clear (and a strength of this study), but the authors do not report on how psychological conditions are diagnosed. How was a diagnosis of anxiety and depression determined? This seems like a major limitation as there is a lot of variability in diagnosing by providers. Also, does this include individuals that see therapists and other mental health or CAMS providers? That is, are these only individuals that was diagnosed by a medical doctor? Does that mean that they received treatment?
2. Was there any missing data and how was missing data handled?
3. Was there any exclusion criteria. Also, why include other health conditions? These other health conditions should be referred to as "comorbid medical conditions" or "other disease" instead of "underlying disease" as underlying suggests they are the cause.
Results: Again very well written and organized. One strength is that UC and CD analyzed separately.
Pg. 13 first paragraph typo - indented on line 250.
Line 250 - replace "needed" with "prescribed"
One general comment: The inclusion of anxiety/depression at baseline is problematic. We do not know if the onset is after diagnosis, corresponding with diagnosis, or before diagnosis but discovered when they saw their doctor for this medical event. Also, the time of medical diagnosis is stressful. Up to 80% of patients will have anxiety at time of diagnosis. It is difficult to differentiate the difference between a natural stress response and clinical levels of anxiety and depression at diagnosis. This should be acknowledge at least in the limitations.
Note: Statistics beyond my expertise, please have a statistician review.
Discussion: This section requires the most significant changes.
Line 297 - place "seriously" before anxious
Lines 305 -315 - the statement starting with, "However, the relative risk for newly..." The authors provide research (citations 24-26) that are in contrast to the findings. The authors need to acknowledge this discrepancy and provide an explanation for the differing findings. Why did these factors not show an increased risk for internalizing symptoms when the literature has shown that they are related. Is there someone in your study design that is different? Is there any literature that may inform the reader as to whey this phenomena was observed?
Lines 315-321 - The explanation of gender differences is not adequately explained. The finding was that males reported more depression. Typically research shows females do. Why would this be? Are there cultural differences? The explanation that it may be heritable in woman does not relate to this finding nor explain this finding.
Line 325 - What does "disease activity of IBD in the baseline" mean? Does the author mean onset or at time of diagnosis? Please clarify.
Line 327 - revise disease "flare up" to "exacerbation". The term flare up is not used in the field anymore
Citations 35-37 do not support the statement that the authors make. It is controversial and difficult to test whether anxiety and depression lead to the onset of IBD. There are several studies showing no relationship too. However, this data does supports that anxiety and depression increases risk for future disease relapse. Please review and describe the literature more precisely.
Limitations need more expansion -- I have provided some thoughts above.
Conclusion/discussion. What is the big picture? Seems like anxiety and depression important to assess within the first year as this is the greatest risk and there is potential impact on disease course. The authors may want to expand on this in the discussion section.
Reviewer 2 Report
The study by Choi et al looks at the prevalence and incidence of anxiety and depression in Korean patients with IBD. The use of a national claims database allows for a population based approach but has several limitations. The study has some novel aspects abut findings overall are not surprising. I have the following points to make:
1. Claims based approaches only find case of anxiety and depression that have been coded / claimed for. In contrast to the use of validated questionnaires / psychological evaluation this approach will underestimate the true incidence.
2. This is ok for using relative risk or hazard ratios but the absolute risk derived will be of little clinical use. The reported incidences are lot lower than in validated questionnaire based studies.
3. I don’t understand why underlying comorbidities were used and why diabetes, hypertension and dislipidaemia were chosen. This makes little clinical sense.
4. The authors show that the IBD codes are validated but this is not mentioned for the anxiety and depression codes.
5. There is a problem with the matching. This has failed as the difference in residence and underlying disease show a significant difference between cases and controls.
6. It is very unusual to have 73% male patients in a disease with usually equal sex distribution.
7. I don’t understand how the use of immunomodulators, steroids and biologics was analysed. It looks like this was used when comparing IBD to controls, which would make no sense. The only way for this to make sense would be to compare IBD patients exposed to immunomodulators, steroids and biologics to those who were not.
8. The results with regards to comorbidities are described in a way that I struggle to follow. To many double negatives, etc.
9. It appears that steroid exposure was associated with less depression and anxiety. This is counterintuitive unless the other patients had uncontrolled disease. In the absence of disease activity data this all makes little clinical sense.
Round 2
Reviewer 2 Report
Kim et al present an updated manuscript that has been considerably strengthened and improved. I am grateful to the authors for the extra analyses undertaken and believe the new data help us in better understanding the data. I still have a few largely minor points to make:
The comorbid conditions can indeed be confounders. The authors should a reference why these may be confounders.
The associations with medication use could either reflect the effects of the medicines used or the underlying disease activity leading to medication use. The current data can not entangle this. Please add a sentence to the discussion highlighting this.
The matching has led to controls that significantly differ. The authors acknowledge that in their response. However this is not sufficiently acknowledged in the study limitations in the discussion.
Page 2 line 60 remove ‘been’
Page 3 line 99 remove ‘causative’
